# Patterns of Cyclist and Pedestrian Street Crossing Behavior and Safety on an Urban Greenway

**DOI:** 10.3390/ijerph16020201

**Published:** 2019-01-12

**Authors:** Christopher E Anderson, Amanda Zimmerman, Skylar Lewis, John Marmion, Jeanette Gustat

**Affiliations:** 1Department of Epidemiology, Tulane University School of Public Health and Tropical Medicine, New Orleans, LA 70112, USA; cander5@tulane.edu (C.E.A.); azimmer2@tulane.edu (A.Z.); 2Center for Aging, Tulane University School of Medicine, New Orleans, LA 70112, USA; slewis17@tulane.edu; 3Department of Global Community Health and Behavioral Sciences, Tulane University School of Public Health and Tropical Medicine, New Orleans, LA 70112, USA; imarmion@tulane.edu; 4Prevention Research Center, Tulane University School of Public Health and Tropical Medicine, New Orleans, LA 70112, USA

**Keywords:** greenway, rectangular rapid flash beacon, pedestrian safety, cyclist safety, motor vehicle yielding

## Abstract

Greenways are linear open spaces, which are often used as trails for pedestrians and cyclists, but junctions with roads are a safety concern and act as a potential impediment to active transportation. This study evaluated crossing behavior patterns and safety at greenway–road junctions in New Orleans, LA. Crossing behaviors, safety and motor vehicle behavior were collected using direct observation methods. Intercept surveys were conducted to assess greenway use and safety perceptions. Logistic and negative binomial regression were used to assess the relationships between crossing signal (rectangular rapid flash beacon) activation and motor vehicle behavior. Fewer unsafe crossings occurred when the crossing signals were activated for cyclists and pedestrians (*p*-values of 0.001 and 0.01, respectively). There was no association between pedestrian use of crossing signals and motor vehicle stopping behavior but cyclists had significantly higher odds of motor vehicles failing to stop when the signal was activated (OR 5.12, 95% CI 2.86–9.16). The activation of rectangular rapid flash beacons at urban greenway junctions with roads did not influence motor vehicle behavior. Differences in crossing safety by signal use cannot be attributed to the signal’s influence on motor vehicle stopping behavior.

## 1. Introduction

Physical activity is an important behavior that promotes health, with increased physical activity being associated with decreased cardiovascular disease, diabetes, cancer and mental health disorders [1,2,3,4]. However, the majority of adults in the United States do not currently achieve the recommended levels of physical activity [5].

Efforts at promoting physical activity have produced only marginal improvements in population level activity [6], which has led to the development of theories of physical activity behavior to inform future promotion activities [7,8]. Guided by the concept of behavior settings, identifying contexts in which individuals choose between active and inactive behaviors, such as transportation, is important for understanding how environmental factors can influence physical activity behavior [7]. Understanding these contexts will enhance physical activity promotion efforts and help to address the population health burden of insufficient physical activity [9]. Transportation is a part of the daily lives of the majority of non-housebound US adults, but most of this transportation occurs by passive means [10]. Transportation may be an area of daily life that is highly amenable to the promotion of physical activity. Furthermore, due to its ubiquity in daily routines, it may represent a target for efficiently addressing the prevalence of insufficient physical activity in adults [11,12].

Appropriate facilities, including sidewalks for pedestrians, on-road bike lanes for cyclists and off-road trails for pedestrians and cyclists, such as those commonly included in greenways [13,14], are important for the promotion of active transportation [15,16,17]. Individuals engaging in active transport may be more susceptible to threats from motor vehicles and other safety concerns compared to those using a motor vehicle for transportation, with risks of death higher both per trip and per mile traveled (23 and 12 times higher than for vehicle occupants per mile traveled for pedestrians and cyclists, respectively) [9]. Pedestrians account for 22–58% of all traffic fatalities in large cities [18] and over 70,000 pedestrians were injured in the US in 2010 [19]. Junctions with motor vehicle traffic represent one of the greatest risks to cyclists and pedestrians [20]. Complete streets policies aim to design transportation systems to accommodate all users, including motor vehicles, pedestrians and cyclists [21]. There has been a rapid expansion in the number of communities with complete streets policies [21], with the likelihood of implementing a policy in a community related to the state obesity and active commuting rates as well as whether adjacent communities have complete streets policies [22]. An evaluation of a Complete Streets statute in Florida found a significant association between the statute and reduced pedestrian fatalities [23]. In light of these policy shifts and the increases in numbers of pedestrians and cyclists, various elements of transportation infrastructure need to be rigorously evaluated to optimize the safety of the increasing number of active commuters.

This study evaluated the safety of major road crossings along an urban greenway in New Orleans, LA in response to concerns reported to community stakeholders. The aim of this study was to observe and describe the patterns of street crossing and vehicle stopping behavior at major streets intersecting the greenway, with a focus on the safety of these crossings, and to quantify concerns and safety issues reported by greenway users. This study evaluated the proportion of crossings that were unsafe, crossing signal activation during attempted crossings, motor vehicle behavior in response to signal activation in addition to the safety concerns and crossing experiences of greenway users. Our hypothesis is that signal use is associated with a lower proportion of unsafe crossings and with a higher proportion of motor vehicles stopping for greenway users at the street crossings.

## 2. Materials and Methods

This study of crossing safety along a pedestrian and cyclist trail in New Orleans involved the systematic observation of crossings at intersections of the greenway and major streets on 24 March 2017 and 1 December 2017. Intercept surveys were conducted during a two-week period in July and August of 2017. All research activities were approved by the Institutional Review Board of Tulane University (1026567-OTH).

### 2.1. Setting

The greenway in this study is a 2.6-mile bicycle and pedestrian trail connecting diverse neighborhoods with multiple commercial and historic destinations, including the French Quarter in New Orleans, LA (Figure 1). The greenway also connects to another off-road bike and pedestrian trail. It was conceived of as a multi-use transportation corridor and linear park [24]. The city invested 9.1 million dollars in the project, breaking ground in 2014 and opening the greenway in 2015. The trail is lit with LED trail lighting and includes curb extensions, signalized high visibility crosswalks and Americans with Disabilities Act-compliant curb ramps at sidewalk corners. Major street crossings are outfitted with rectangular rapid flash beacons (RRFB), which are user-actuated amber light emitting diodes (LEDs) that can be actively or passively triggered [25]. One of the crossings included in this study is shown (Figure 2) from the perspectives of the motor vehicle (Figure 2A) and greenway user (Figure 2B). Signs and beacons are marked with red arrows and the location of crossing on the road is circled in red in Figure 2A. The RRFBs in the present study require manual activation with a push button. The greenway averages 830 daily total users and the majority are thought to be cyclists but the counts are from infrared sensors and cannot be separated by user type [26]. Louisiana is ranked third among all states for the percent of traffic fatalities that are cyclists [27]. Two streets (Claiborne, Carrollton) included in this sample are among the most dangerous pedestrian corridors in New Orleans and another street (Broad) contains one of the most dangerous intersections for pedestrians in the city [28].

### 2.2. Crossing Observations

Trained research assistants were assigned to teams that were stationed at the intersections of the greenway and nine major intersecting streets to observe the crossing behavior of greenway users and motor vehicle behavior at the crossings. Direct observations were conducted at all intersections along the greenway equipped with a RRFB. Weather conditions were similar on both days of observation, with clear to partly cloudy skies and no precipitation during the observation period. Motor vehicle behavior was only recorded while the crossings were in use. Observations on 24 March 2017 were collected to assess the safety of crossings when the RRFBs were and were not activated. One research assistant at each intersection was assigned to observe pedestrians, one to observe cyclists and one to observe motor vehicles. All observers began and stopped recording simultaneously. All intersections were observed for 150 minutes. Observations on 1 December 2017 were collected to assess whether motor vehicles responded differently to attempted crossings by greenway users when the RRFBs were and were not activated. One research assistant at each intersection was assigned to observe motor vehicle behavior in each lane of traffic and the final observer for each intersection was assigned to call the signal. Observers were assigned specific locations at the intersection to enhance their view of the intersection while minimizing the visibility of observers from the road or the greenway. This was done in an attempt to diminish the impact that the presence of observers may have on motor vehicle or greenway user behavior. A total of 32.5 hours of crossing observations was collected across all intersections and both days of direct observation.

Pedestrians and cyclists crossing the intersection were recorded as having activated or not activated the RRFB and further recorded as having made a correct, errant, mistaken or unsafe crossing. A correct crossing was defined as a pedestrian or cyclist crossing through the entire intersection within the lines of the crossing without any errant behavior. An errant crossing was defined as a pedestrian or cyclist crossing the intersection diagonally, either beginning or ending the crossing outside the marked crosswalk. A mistaken crossing was a pedestrian or cyclist who aborted a crossing and returned to the original position. An unsafe crossing was defined as a pedestrian or cyclist who entered the crossing without regard for traffic or entered the road crossing part-way before stopping because traffic did not allow them to complete the crossing.

Vehicle behavior was recorded only when there was a pedestrian or cyclist attempting to cross the street. Motor vehicles were recorded as having made a correct stop, an incorrect stop, failing to stop, failing to stop when the adjacent lane was stopped or having conflict with a greenway user. A correct stop was defined as a motor vehicle coming to a complete stop behind the designated white line on the road, allowing the pedestrian or cyclist to cross the intersection fully before resuming forward progress. An incorrect stop was recorded if the motor vehicle stopped past the white line and possibly obstructed the crossing. A conflict was recorded if a vehicle hit a pedestrian, sped up, stopped short, swerved or performed evasive or aggressive actions or if the pedestrian had to make similar, sudden movements in an attempt to avoid being hit. A vehicle that did not stop for the greenway user was recorded as failing to stop.

Duplicate observations were collected to examine reliability for motor vehicle observers and signal callers at all intersections on 1 December. Duplicate observations for signal caller data were collected for 17% of crossings, and duplicate observations for motor vehicle data were collected for 18% of lane specific observed crossings. Kappa was used to assess reliability, which was found to be high (0.87 and 0.94 for motor vehicle and signal caller observations, respectively).

### 2.3. Intercept Survey

Intercept surveys were conducted over a two-week period in July and August of 2017. Trained interviewers, in teams of two, visited locations along the greenway on different days and at different times and approached individuals using the greenway. Greenway users of at least 18 years of age were asked if they would consent to an interview. Refusals were tallied and reasons for refusal were recorded. Individuals who consented to participate were asked questions about their use of the greenway, perceived safety, the RRFB equipped crossings and to mention any other concerns that were not addressed in the structured portion of the interview. Addresses of residence were geocoded using ArcGIS 10.3 (Environmental Systems Research Institute, Redlands, CA, USA) and were used to identify neighborhoods of residence and distance to the nearest greenway access point for respondents. Neighborhoods around greenway–road junctions in this study were characterized with data from the American Community Survey [29].

### 2.4. Statistical Methods

Data from the crossing observations were tallied and the proportion of crossings in each behavior category was calculated by and across intersections. Differences in proportions were evaluated using Fisher’s exact test. Correlations between the proportion of cyclists and pedestrians who activated the RRFB and the proportion of motor vehicles in different categories of vehicle behavior in each period of observation were determined using Spearman’s rank correlation coefficient.

Motor vehicle behavior was summarized by calculating a score (possible range 1–4) for each crossing, lane of traffic and intersection. Each failure to stop contributed 4 points, each conflict contributed 3 points, each obstruction contributed 2 points, each stop past the white line contributed 2 points and each correct stop contributed 1 point. The total points in a crossing period were divided by the number of motor vehicles observed to give a score between 1 and 4, with higher scores indicating that the average motor vehicle behavior at the crossing was more dangerous.

Descriptive statistics were calculated for the data collected during the intercept interviews. Cyclist and pedestrian responses were compared for ease of crossing intersections with RRFBs, frequency of RRFB activation, reasons for not activating the RRFBs as well as issues with vehicles, cyclists and pedestrians along the greenway.

The relationship between greenway-user RRFB activation and motor vehicle behavior at the crossings was evaluated by testing the proportions of motor vehicles in each lane of traffic in each category of behavior between RRFB-activated and RRFB-not-activated crossings with t-tests. Additionally, the relationship between RRFB activation and whether at least one motor vehicle in a given lane of traffic failed to stop was evaluated with chi-square and Fisher’s exact tests for cyclists and pedestrians, respectively. As any failure to stop by a motor vehicle during an attempted crossing represents a significant safety hazard to pedestrians and cyclists attempting to cross lanes of traffic, associations between RRFB activation and the failure of any motor vehicle to stop in a lane of traffic were assessed using logistic regression models [30]. Logistic regression of a dichotomous outcome does not make full use of the information contained in the count data so associations between RRFB activation and the count of vehicles that failed to stop in each lane of traffic were assessed using negative binomial regression models [31,32]. Adjustments were made in logistic and negative binomial models, including adjusting for whether a cyclist or pedestrian was using the crossing, an interaction between greenway user-type and RRFB activation (model 1), model 1 variables and motor vehicle behavior scores for the lane of traffic and street under observation (model 2), and model 2 variables and indicators of time to accommodate potential linear and non-linear period effects (model 3). Clustering within crossings, within the lane of traffic and within the street was explored using generalized estimating equations (GEE) for both logistic and negative binomial regression models.

All analyses were conducted using SAS 9.4 (SAS Institute Inc., Cary, NC, USA).

## 3. Results

Characteristics of the intersections included in the study sample and the surrounding neighborhoods are presented in Table 1. A total of 861 cyclists, 318 pedestrians and 923 motor vehicles was recorded during direct observations of street crossings. The RRFBs were activated by a small percentage of cyclists and pedestrians on both days of crossing observations (10.5% and 22.8% for cyclists and pedestrians, respectively on 24 March; 15.5% and 22.2% for cyclists and pedestrians, respectively on 1 December). The association between crossing type and RRFB activation is shown in Table 2. Crossings made while the RRFBs were not activated were more likely to be unsafe for cyclists and pedestrians (*p*-values of 0.001 and 0.01, respectively).

The association between RRFB activation and motor vehicle behavior at greenway crossings was assessed (data not shown). In the data collected on 24 March, no statistically significant correlation was observed between the proportion of individuals who activated the RRFBs during a crossing period and the proportion of motor vehicles in any category of behavior. There was no correlation between the summary motor vehicle behavior score (2.49 ± 1.70 (mean ± standard deviation)) and proportion of individuals who activated the RRFBs. There were no significant differences in the proportion of motor vehicles in each category of motor vehicle behavior when comparing the data collected on 1 December when the RRFB was activated and not activated. The proportion of vehicles making any stop was higher when the RRFB was not activated but this did not achieve statistical significance.

The majority of the 122 respondents to the intercept surveys were cyclists (Table 3). Seventy-four percent of the individuals approached refused to participate (93% of refusals indicated that they did not have time to complete the survey). Twenty-nine percent of respondents were female, 77% were white and the median age of respondents was 40 (interquartile range 29, 49). A higher percent of cyclists reported using the greenway for transportation to work or school than pedestrians while a higher percent of pedestrians reported using the greenway for exercise. Pedestrians reported significantly more RRFB activation (*p*-value of 0.01). Both groups cited not needing the RRFBs and motor vehicles not stopping for the RRFBs as the primary reasons for not using them. More cyclists reported prior conflict(s) with a motor vehicle that required evasive action (54% vs. 24%, *p*-value of 0.01). A higher proportion of pedestrians than cyclists reported that cars stop when the RRFBs are activated (55% vs. 39%) but this did not achieve statistical significance. The majority of survey respondents lived in neighborhoods along the greenway (Figure 1).

The association between RRFB activation and the odds of a vehicle failing to stop during an attempted crossing was modified by whether the individual attempting the crossing was a cyclist or a pedestrian. Logistic regression models for whether a motor vehicle failed to stop in a lane of traffic during a given crossing episode found no association for RRFB activation in pedestrians (OR 1.04, 95% CI 0.37–2.91). However, the odds of any motor vehicle failing to stop for a cyclist were five times higher when RRFBs were activated than when RRFBs were not activated (OR 5.12, 95% CI 2.86–9.16) (Table 4). These results were not attenuated by statistical adjustments or by accommodating the clustering within lanes of traffic and street or crossing episode and street.

Negative binomial regression models were run for the count of motor vehicles in a lane of traffic failing to stop during a given crossing episode and the results are presented in Table 5. RRFB activation was not associated with the number of motor vehicles failing to stop for pedestrians but was associated with a significant increase in the number of motor vehicles failing to stop for cyclists, with an additional 1.48 motor vehicles failing to stop for cyclists who activated the RRFB compared to those that did not (*p*-value < 0.0001). This association did not change after adjustment for the mean motor vehicle behavior scores for the street and lane of traffic or after adjustment for potential period effects. The results of generalized estimating equation models that considered the clustering of observations within lane of traffic and street as well as clustering within the crossing episode and street were similar to those from models with no clustering.

## 4. Discussion

Most cyclists and pedestrians using the greenway did not activate the RRFBs at street crossings. Fifty-seven percent of greenway users interviewed in this study reported that they activated the RRFBs less than they otherwise would to cross streets because they believed that motor vehicles do not stop for the beacons. The inverse association observed between RRFB activation and unsafe crossings was not reflected in an association between RRFB activation and motor vehicle stopping behavior on either day of crossing observation. These results might suggest that the RRFBs do not make crossings safer by altering motor vehicle behavior but instead indicate that the pedestrians and cyclists who use the signals are more cautious.

RRFBs received interim approval from the Manual of Uniform Traffic Control Devices in 2018 [33]. Prior research on the impact of RRFBs on motor vehicle behavior has found that RRFBs generally increase the frequency of motor vehicles yielding to pedestrians, with the introduction of RRFBs being associated with increased driver yielding in a variety of settings [34,35,36,37]. Most of these studies were conducted on roads with speed limits of 35 mph, which is the speed limit on all streets included in the present study sample. Our study results do not support the findings of an evaluation conducted in Southern California, which found that the proportion of drivers yielding to pedestrians generally increased when RRFBs were activated [38] although the researchers themselves activated the signals. Furthermore, that evaluation found that RRFB installations were associated with a decrease in drivers yielding during the dusk evaluation. The results also contrast with an evaluation of a RRFB installed along a trail in Florida, which demonstrated significant improvement in motor vehicle yielding to cyclists and pedestrians after RRFB installation. Furthermore, this study found a significant increase in motor vehicle yielding when the RRFB was activated (51% of crossings) compared to when it was not activated (54% yield compared to 14%) [39]. In a more recent quasi-experimental study, neither RRFB presence nor activation (signals were automatically activated for every crossing) was associated with increased driver yielding [40]. Signal activation rates were substantially lower in this study compared to other observational studies in which signal activation rates exceeded 50% [35,39].

Our results indicate that the RRFBs at locations where the greenway crosses motor vehicle traffic do not work as intended for cyclists and pedestrians in this high traffic urban setting. The absence of improved yielding behavior among motor vehicles in response to RRFBs along the greenway in this study conflicts with prior data evaluating driver yielding before and after the introduction of RRFBs [36,37]. The evidence from the evaluations of the influence of RRFB signal activation on driver yielding has been less consistent [38,39,40] and our results are consistent with those from an evaluation of RRFB treatments on a college campus in Virginia [40].

Signal activation was not associated with a decrease in the number of motor vehicles that failed to stop for a pedestrian and was unexpectedly associated with an increase in both the odds of any vehicle failing to stop and the number of motor vehicles that failed to stop for cyclists. It is unlikely that motor vehicles respond differently to the signal based upon the type of greenway user that activated it. This may indicate that cyclists and pedestrians use the signal differently. Activating the RRFB may not be a burden for pedestrians, who may activate the beacons consistently when attempting to cross roads with substantial motor vehicle traffic. Cyclists, for whom slowing or stopping to press a button to activate the beacon possibly represents a significant inconvenience, may only utilize the signal when they must come to a stop due to heavy motor vehicle traffic on the street [39].

This study has several strengths that merit note. Data were collected on motor vehicle and crossing behaviors to allow the evaluation of associations of the activation of RRFBs with both the safety of the crossing and motor vehicle yielding behavior. All intersections with RRFBs along the greenway were included in the sample. Data were collected at all intersections simultaneously, which should limit confounding introduced by variables that may vary over time. Duplicate observations were captured for a substantial proportion of crossings, which allowed for the assessment of the reliability of the motor vehicle and crossing behavior observations. Intercept surveys were conducted to assess the perspective of individuals who use the greenway, allowing the assessment of both objective and perceived measures of crossing safety and RRFB function. There are also limitations that should be mentioned. There was no assessment of crossing safety and motor vehicle yielding at these intersections before the RRFBs were installed so the comparison of motor vehicle yielding performed in this analysis between when the beacons were and were not activated is the most comprehensive analysis possible with the data. The presence of observers at the crossings may have altered motor vehicle or greenway user behavior. Finally, we were unable to assess the driver’s perspective regarding the greenway crossings and the RRFB signals.

Infrastructure for cycling is important to promote active transportation and cycling for exercise purposes [17]. Safety concerns represent a significant barrier to increasing ridership [41], particularly among the substantial proportion of the population who are interested in cycling but apprehensive about potential safety issues [42,43,44]. Prior research has demonstrated improved cyclist safety when bicycle paths are available [45]. Our results demonstrate that street crossings represent a major safety concern for users of off-road trails such as the greenway in this study. Our findings demonstrate that nearly six percent of cyclist crossings were unsafe. Assuming that 730 of the 830 daily users are cyclists [26], if each cyclist crosses only 1 intersection with a RRFB per day, that would correspond to 44 unsafe crossings each day. If each cyclist crosses all 9 intersections with a RRFB, there would be nearly 400 unsafe crossings each day. Each of these unsafe crossings represents an opportunity for a serious injury or death of a greenway user during a collision with a motor vehicle. The absence of association between RRFB use and increased motor vehicle yielding at the crossings indicates that additional approaches are needed to enhance the safety of crossings in this setting.

## 5. Conclusions

Prior research has demonstrated that RRFBs can improve motor vehicle yielding to pedestrians and cyclists, but those improvements have not been consistently observed in the literature. The activation of RRFBs installed at major street crossings along an urban greenway in New Orleans, LA did not have an impact on motor vehicle behavior. Further research is needed to assess whether signals, such as RRFBs, function as intended to increase motor vehicle yielding in other settings and may benefit from including crash-based evaluation. In this setting, further means of promoting motor vehicle yielding to cyclists and pedestrians are needed. These methods could include promoting signal activation among cyclists and pedestrians, changing how the beacon is activated to make it easier for cyclists to activate them, educating motorists about how and when to respond to the beacons, installing advanced warning signs about the crossing, enforcing who has right of way, installing traffic calming features, such as speed bumps, or installing other types of signals. Future studies should evaluate changes in crossing safety before and after the installation of new crossing signals as well as comparing crossing safety when the signals are and are not in use.

## Figures and Tables

**Figure 1 ijerph-16-00201-f001:**
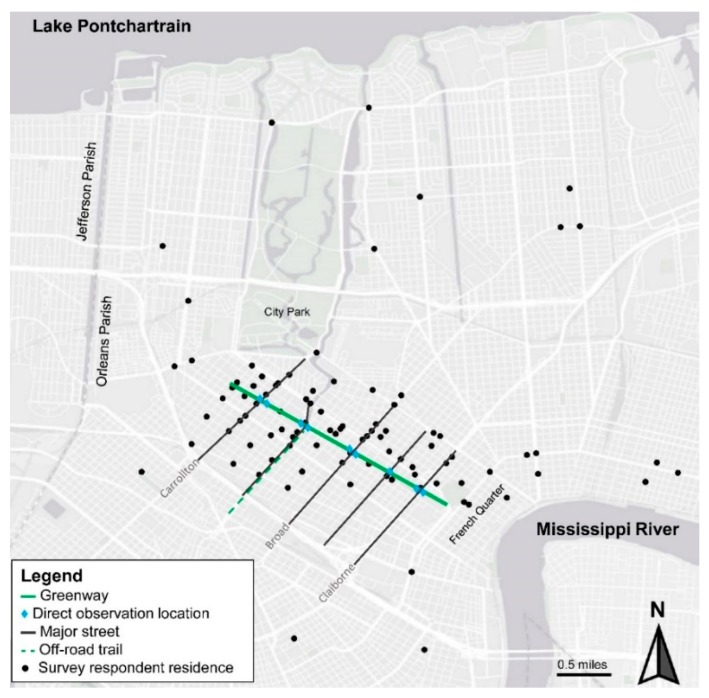
Map of the greenway and residences of intercept survey respondents in Orleans Parish, Louisiana.

**Figure 2 ijerph-16-00201-f002:**
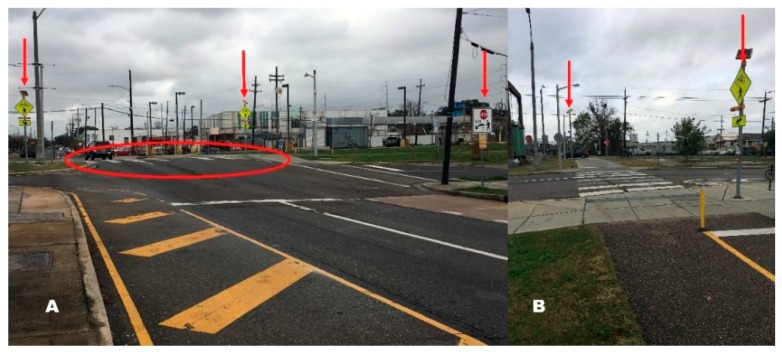
Crossing equipped with rectangular rapid flash beacon as seen from the perspective of (**A**) motor vehicle user and (**B**) greenway user.

**Table 1 ijerph-16-00201-t001:** Characteristics of street crossing locations included in this study.

Intersection	Neighborhood Characteristics †							Street Characteristics ††		
	Commute									
	Median Income	White (%)	Own Home (%)	No Car Access (%)	Walk (%)	Bicycle (%)	Public Transit (%)	Lanes	Mean Motor Vehicle Behavior Score *	Average Daily Traffic **
1	36,548	75	33	18	1	8	7	3	1.90	13,172
2								3	2.59	25,949
3	34,808	38	35	25	2	6	6	2	1.82	12,949
4								2	2.74	15,571
5								2	2.37	42,951
6								2	3.07	33,604
7	17,244	15	24	45	15	6	15	4	3.52	5136
8								2	2.78	37,145
9								2	3.50	16,873

† The neighborhood was defined by the census tract in which the intersection was located and characterized using American Community Survey 2011–2016 estimates. For intersections 7, 8 and 9, two adjacent tracts were combined because the tract in which the intersections are located had too few residents (*n* = 158) to report median income. †† All streets had medians, a speed limit of 35 miles per hour, crosswalk markings and a yield to pedestrian sign (MUTCD R1–6) in addition to the rectangular rapid flash beacon. * Mean motor vehicle behavior score was calculated as the average of motor vehicle scores (for each crossing) for each intersection. Scores were calculated by adding 4 points for each motor vehicle that failed to stop, 3 points for each conflict with a greenway user, 2 points for incorrect stopping behaviors and 1 point for correct stops. The total score was then divided by the number of vehicles observed to give a score between 1 (all cars stopped correctly) and 4 (all cars failed to stop). Higher scores indicate that the average motor vehicle behavior at crossings was more dangerous. ** Most recent average daily traffic count provided by New Orleans Regional Planning Commission. These data were from 2014 to 2017 for all streets, excluding 3, 7 and 8, for which the most recent data was from 2008.

**Table 2 ijerph-16-00201-t002:** Crossing characteristics by RRFB activation for pedestrians and cyclists.

	**Pedestrian**				***p*-Value ***
	**RRFB Activated**		**RRFB not Activated**		
	***n* = 55**		***n* = 186**		
	***n***	**%**	***n***	**%**	
Crossing Type †					0.01
Correct	50	90.91	138	74.19	
Errant	1	1.82	30	16.13	
Mistaken	0	0.00	1	0.54	
Unsafe	4	7.27	17	9.14	
	**Cyclist**				***p*-Value ***
	**RRFB Activated**		**RRFB not Activated**		
	***n* = 62**		***n* = 528**		
	***n***	**%**	***n***	**%**	
Crossing Type †					0.001
Correct	61	98.39	432	81.82	
Errant	1	1.61	61	11.55	
Mistaken	0	0.00	0	0.00	
Unsafe	0	0.00	35	6.63	

RRFB, rectangular rapid flash beacon. † Correct: a pedestrian or cyclist crossing through the entire intersection within the lines of the crossing, without any errant behavior. Errant: a pedestrian or cyclist crossing the intersection diagonally, either beginning or ending the crossing outside the marked crosswalk. Mistaken: a pedestrian or cyclist aborted a crossing and returned to the original position. Unsafe: a pedestrian or cyclist entered the crossing without regard for traffic, or entered the road crossing part-way before stopping because traffic did not allow them to complete the crossing. * *p*-values from Fisher’s exact test.

**Table 3 ijerph-16-00201-t003:** Safety concerns and barriers to RRFB equipped crossing use among respondents to intercept surveys.

Greenway User Survey Response	Cyclist		Pedestrian		*p*-Value
	*n* = 93		*n* = 29		
	*n*	%	*n*	%	
Purpose of Greenway use					
Get to work/school	56	60.2	13	44.8	0.1
Get exercise	38	40.9	20	69.0	0.008
Frequency of Greenway use					0.6
Daily	55	59.1	20	69.0	
Weekly	24	25.8	7	24.1	
Less than weekly	14	15.1	2	6.9	
How often do you activate RRFBs?					0.01
Always/Almost always	27	29.0	17	58.6	
Sometimes	33	35.5	10	34.5	
Rarely/Never	33	35.5	2	6.9	
Reason for not activating RRFBs?					
Location is inconvenient	21	22.6	2	6.9	0.06
Don’t need it	67	72.0	18	62.1	0.3
Drivers don’t stop for RRFB	56	60.2	13	44.8	0.1
Feel that drivers at crossings					
speed up	24	25.8	14	48.3	0.07
skid/swerve	18	19.4	5	17.2	0.4
stop in crosswalk	29	31.2	7	24.1	0.5
behave well	66	71.0	25	86.2	0.11
In prior crossings					
cars kept going	86	92.5	27	93.1	0.9
cars necessitated evasive action	50	53.8	7	24.1	0.01
cars stopped short	33	35.5	11	37.9	0.8
cars skidded	13	14.0	4	13.8	0.9
cars blocked crosswalk	37	39.8	7	24.1	0.1
There is good visibility at crossing	84	90.3	28	96.6	0.4
The crossing signals make cars stop	36	38.7	16	55.2	0.07

RRFB, rectangular rapid flash beacon.

**Table 4 ijerph-16-00201-t004:** Association between RRFB activation and failure of any motor vehicle to stop in each lane of traffic at greenway street crossings in logistic regression models.

RRFB Activation	Model 1		Model 2		Model 3	
	OR	95% CI	OR	95% CI	OR	95% CI
No Clustering						
Cyclist	5.12	2.86–9.16	5.49	3.01–10.01	5.63	3.07–10.36
Pedestrian	1.04	0.37–2.91	1.13	0.40–3.19	1.08	0.38–3.10
Clustering Within Lane, Street						
Cyclist	5.05	2.41–10.57	5.52	2.57–11.86	5.70	2.70–12.05
Pedestrian	1.05	0.44–2.52	1.13	0.46–2.75	1.09	0.43–2.73
Clustering Within Crossing, Street						
Cyclist	5.36	2.79–10.31	5.76	3.01–10.99	5.88	3.06–11.30
Pedestrian	1.06	0.43–2.59	1.14	0.46–2.82	1.10	0.44–2.74

RRFB, rectangular rapid flash beacon. CI, confidence interval. OR, odds ratio. Model 1: Unadjusted. Model 2: Adjusted for street and lane motor vehicle behavior scores. Model 3: Adjusted for street and lane motor vehicle behavior score, with period indicators used to accommodate linear and quadratic period effects.

**Table 5 ijerph-16-00201-t005:** Association between RRFB activation and count of motor vehicles failing to stop in each lane of traffic at greenway street crossings in negative binomial regression models.

RRFB Activation	Model 1			Model 2			Model 3		
	β	SE	*p*-Value	β	SE	*p*-Value	β	SE	*p*-Value
No Clustering									
Cyclist	1.483	0.194	<0.0001	1.468	0.205	<0.0001	1.430	0.213	<0.0001
Pedestrian	−0.037	0.434	0.93	0.028	0.435	0.95	−0.034	0.438	0.94
Clustering Within Lane, Street									
Cyclist	1.408	0.199	<0.0001	1.470	0.210	<0.0001	1.442	0.204	<0.0001
Pedestrian	−0.023	0.340	0.95	0.028	0.370	0.94	−0.033	0.385	0.93
Clustering Within Crossing, Street									
Cyclist	1.515	0.219	<0.0001	1.503	0.206	<0.0001	1.464	0.214	<0.0001
Pedestrian	−0.028	0.344	0.94	0.026	0.343	0.94	−0.033	0.343	0.92

RRFB, rectangular rapid flash beacon. SE, standard error. Model 1: Unadjusted. Model 2: Adjusted for street and lane motor vehicle behavior scores. Model 3: Adjusted for street and lane motor vehicle behavior score, with period indicators to accommodate linear and quadratic period effects.

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
