# Peer review of "Patterns of Cyclist and Pedestrian Street Crossing Behavior and Safety on an Urban Greenway"

_ijerph, 2019, doi:10.3390/ijerph16020201_

Round 1
Reviewer 1 Report
The manuscript is interesting and properly constructed and written, however at a first glance the presented hypothesis seems to be obvious, so one might hesitate why to study a common sense knowledge? However, the presented results do not prove it (activating of RRFBs does not have an impact on vehicles behaviour). At the same time, the authors mention other studies showing different results. So, it'd be intersting to further explain what had influenced the results - is it the case of location and speciality of the case study area, or maybe the research should be designed differently, e.g. other samples, or other methodology of analyses?
More detailed comments are presented below.
The introduction includes a profound review of relationships between traffic facilities, safety, transport patterns in US, etc., but there is very little information about greenways and their charactristics and roles. As in lines 22-23 - greenways refer to a braod term, so it sounds as generalisation.
Materials and Methods
The graphic presentation of a study area is not clear. There should be a map showing both the greenway, the crossings, the major streets, places with signals, etc. In Fig. 1 there is no scale! From that map, you can't understand what the greenway is - is it a path with accompanyning street greenry or a linear green space? Why the greenway is not linked to other paths? It's important to show the land-use analysis of the area (e.g. you talk about the French Quarter, but the redears don't know where is it). the figure dosn't show any connection of the studied greenway. Maybe you can add photographs of greenway design solutions (lines 90-95).
There is no information on traffic intensity - both in terms of vehicles on major streets and pedestrians / cyclists using the greenway. Is it a popular path used for a transit movement?
Lines 83-84: I doubt if this information here is necessary. As far as I understand the performed research has no ethical restrictions.
Paragraph 2.2
- crossing were the observations were conducted should be marked on the map
- 150 minutes of observations - is it enough?
- why the 2 dates of research observations were so distant (almost 9 months)
- line 135 - did it really happen that a car hit a pedestrian?
lines 141-142 - What do you mean that observations for signal callers data were collected for 18% of lana specific crossing? so how many crossing you studied? And how those 18% were selected (lack of criteria)?
Paragraph 2.3
Intercept survey - how many people you approached (size of sample - is it statistically significant or not)?
Paragraph 2.4
How many vehicles were observed in total?
How many cyclists / pedestrians were observed?
Results
In general, the results are well described, but it would be very useful to show the location of studied crossings (it could be added to figure 1). Moreover, there is no explanation how have you selected those street crossings (lack of delimitation criteria).
Author Response
Reviewer 1:
General Comments:
The manuscript is interesting and properly constructed and written, however at a first glance the presented hypothesis seems to be obvious, so one might hesitate why to study a common sense knowledge? However the presented results do not prove it (activating of RRFBs does not have an impact on vehicles behavior). At the same time, the authors mention other studies showing different results. So, it’d be interesting to further explain what had influenced the results – is it the case of location and specialty of the case study area, or maybe the research should be designed differently, e.g. other samples, or other methodology of analyses?
Thank you for your comments. We agree that the presented hypothesis seems like common sense, but we began this study in response to reports to community stakeholders about frequent safety issues with the RRFBs installed along the greenway (introduction, page 2, line 70). We do find it interesting that results about the effectiveness of RRFBs have not been consistent across different studies. The consistency of the results across the two rounds of direct observation, and the multiple analyses done (correlations, logistic and negative binomial regression) suggest that the analytic approach does not explain the observed result. Our study included all greenway-road intersections equipped with a RRFB, and therefore we feel that it is unlikely to be an issue of the sample of intersections but could be due to other local factors such as signal activation patterns, motor vehicle traffic, and other street intersection features. We are unable to provide an answer as to which of these items underlies the lack of influence of RRFB activation on motor vehicle yielding based on the current analysis but believe future evaluations should explore these possibilities (conclusions page 15, lines 367-376).
Specific Comments:
1. The introduction includes a profound review of the relationships between traffic facilities, safety, transport patterns in US, etc., but there is very little information about greenways and their characteristics and roles. As in lines 22-23- greenways refer to a broad term, so it sounds as generalization.
Thank you, we have now provided some clarification of how greenways fit into the catalog of active transportation infrastructure, with two references for more detailed reading for interested readers, on page 2 lines 53-54.
2. The graphic presentation of the study area is not clear. There should be a map showing both the greenways, crossings, the major streets, places with signals, etc. In Fig. 1 there is no scale. From that map you can’t understand what the greenway is- is it a path with accompanying street greenery, or a linear space. Why the greenway is not linked to other paths? It’s important to show the land-use analysis of the area (eg. you talk about the French Quarter, but readers don’t know where it is). The figure doesn’t show any connection of the studied greenway. Maybe you can add photographs of greenway design solutions (lines 90-95).
We have added additional details to the Figure 1 (page 3) of the greenway, including the mentioned areas at each end of the trail, major streets and intersections equipped with rectangular rapid flash beacons which were included in this study.
We have additionally included a Figure 2 (page 4) with images of one the intersections of the greenway with a major cross street, showing the intersection from the perspectives of motor vehicle drivers (Figure 2A) and greenway users (Figure 2B).
3. There is no information on traffic intensity – both in terms of vehicles on major streets and pedestrians/cyclists using the greenway. Is it a popular path used for a transit movement?
The traffic intensity on the streets observed in this study are reported in Table 1 (page 7: average daily traffic counts are provided). This is a popular path for active transportation in the city, with an average of 830 daily users (page 3, line 95 and page 15, lines 356-357).
4. Lines 83-84: I doubt if this information here is necessary. As far as I understand the performed research has no ethical restrictions.
We have provided information on the ethics approval of the study to meet the requirements of this journal, and because our institution requires all research to submit a protocol to achieve either human subjects research exemption or IRB approval.
5. Crossing where the observations were conducted should be marked on the map.
Thank you, we have marked the crossings where observations were conducted on the map in Figure 1 (page 3).
6. 150 minutes of observations- is it enough?
150 minutes was the duration of continuous observation at each intersection for each day of crossing observations. Across the two days of observation, crossing behavior and motor vehicle behavior were recorded for a total of 32.5 hours. We have added the total duration of time during which crossing observations were collected on page 4 lines 123-124. The duration of observation is comparable to a recently published quasi-experimental evaluation of RRFB effectiveness (Porter BE, Neto I, Balk I, Jenkins JK. Investigating the effects of Rectangular Rapid Flash Beacons on pedestrian behavior and driver yielding on 25mph streets: A quasi-experimental field study on a university campus. Transp Res Part F Traffic Psychol Behav. 2016 2016/10/01/;42:509-21,https://www.doi.org/10.1016/j.trf.2016.05.004). While we have fewer total hours of observation, the quasi-experimental study attempted to capture change in motorist behavior over time, and our analysis is cross-sectional in nature.
7. Why the 2 dates of research observations were so distant (almost 9 months).
This separation occurred because following the initial period of observations and intercept interviews, initial results were tabulated and shared with community stakeholders. The second day of direct observation was designed to address additional questions raised by the community stakeholders.
8. Line 135- did it really happen that a car hit a pedestrian?
No, there were no collisions during the period of observation. A conflict was recorded IF a vehicle hit a pedestrian, sped up, stopped short, swerved or performed evasive or aggressive actions or if the pedestrian had to make similar, sudden movements in an attempt to avoid being hit.
9. Lines 141-142 – What do you mean that observations for signal callers data were collected for 18% of lane specific crossings? So how many crossings you studied? And how those 18% were selected (lack of criteria)?
Duplicate observations were collected by gold standard raters in the direct observation methodology. They were collected for all observed crossing episodes while those gold standard raters were at each intersection, but due to the inability of the gold standard raters to be at all 9 intersections simultaneously and transit times between the crossings, not all episodes were observed in duplicate.
A total of 571 crossings were observed were observed on the second day of direct observation.
10. Intercept survey- how many people you approached (size of sample- is it statistically significant or not)?
A number, such as a sample size, cannot be statistically significant. As is now stated in the text (page 8, line 224), 122 interviews were conducted. A total 470 individuals were approached to attempt an interview.
11. How many vehicles were observed in total?
A total of 923 motor vehicles were observed during this study. This has now been mentioned in the results on page 6 lines 197-198.
12. How many cyclists/pedestrians were observed?
A total of 861 cyclists and 318 pedestrians were observed during this study. This has now been mentioned in the results on page 6 lines 197-198.
13. In general, the results are well described, but it would be very useful to show the location of the studied crossings (it could be added to Figure 1). Moreover, there is no explanation of how you have selected those street crossings (lack of delimitation criteria).
Thank you, we have now indicated the locations of the studied crossings in Figure 1 (page 3). All intersections with a RRFB along the trail were included in this study (there were no intersections with RRFBs excluded from the direct observations), which is now mentioned on page 4, lines 110-111.

Reviewer 2 Report
The article deals with the issue of cyclist and pedestrian street crossing behavior and safety on an urban greenway. The findings are astonishing since it appeared the RRFBs did not improve motor vehicle yielding to pedestrians and cyclists. The Authors’ conclusions suggested there is a need to apply some other solutions to force the drivers to behave suitably. The paper is generally well written, however I have a few questions and remarks:
1. Lines 98 – 100. Could the Authors mark these dangerous spots in Figure 1.
2. Lines 104 – 105. The same – could the Authors mark the crossings being observed.
3. Table 1 presets among others the averaged traffic intensity (traffic volume per day). As can be seen, this value varies significantly for particular intersections. Could the Authors check whether the drivers’ behavior depends on traffic intensity?
4. Lines 172 – 186. Although I don’t contest the selection of the statistical methods for the successive tasks I think a short substantiation of such choice would be valuable.
5. What was the total number of intercept surveys?
6. Please explain the difference between both descriptions of scoring of the motor vehicle behavior: lines 162 – 165 (correct stop = 0 points) and lines 201 – 204 (correct stop = 1 point).
Author Response
Reviewer 2:
General Comments:
The article deals with the issue of cyclist and pedestrian street crossing behavior and safety on an urban greenway. The findings are astonishing since it appeared the RRFBs did not improve motor vehicle yielding to pedestrians and cyclists. The Authors’ conclusions suggested there is a need to apply some other solutions to force the drivers to behave suitably. The paper is generally well written, however I have a few questions and remarks
Thank you for your comments and suggestions. We agree that the lack of influence of the RRFBs on motor vehicle yielding to pedestrians and cyclists is striking and hope these results will encourage additional solutions to modify motor vehicle behavior when interacting with pedestrian and cyclists on greenways.
Specific Comments:
1. Lines 98-100. Could the authors mark these dangerous spots in Figure 1.
We have added additional information to Figure 1 (page 3), including the labels of the streets which are dangerous pedestrian corridors or contain the dangerous pedestrian intersections mentioned, and provided additional information in the text (page 3, lines 98-99).
2. Lines 104-105. The same- could the authors mark the crossings being observed.
We have added additional information to Figure 1 (page 3), including the intersections equipped with rectangular rapid flash beacons which were included in this study.
3. Table 1 presents among others the averaged traffic intensity (traffic volume per day). As can be seen, this value varies significantly for particular intersections. Could the authors check whether the drivers’ behavior depends on traffic intensity?
The developed motor vehicle behavior score can be used to compare the motor vehicle behavior between the intersections- motor vehicle behavior wasn’t clearly related to average daily traffic. The Pearson correlation coefficient for Average Daily Traffic and mean Motor Vehicle Behavior Score was -0.08, and the Spearman correlation coefficient for the ranks was -0.02. A positive correlation between Average Daily Traffic and mean Motor Vehicle Behavior Score would indicate that driver behavior is worse when traffic is heavier. These very small negative correlation coefficients indicate that there is no correlation between traffic intensity and drivers’ behavior, however this is limited by the very small sample of 9 intersections. We do not have traffic intensity in smaller units of time than a daily average and are therefore unable to evaluate the relationship between traffic intensity and motor vehicle behavior is related within each street. We therefore feel that this analysis is too limited to merit inclusion in the paper.
4. Lines 172-186. Although I don’t contest the selection of the statistical methods for the successive tasks, I think a short substantiation of such choice would be valuable.
Logistic regression was used to model the relationship between signal activation and the failure to stop of any motor vehicle, because we felt that even one failure to stop reflected a substantial danger to greenway users. This is now mentioned on pages 5-6, lines 181-184.
We used negative binomial regression to model the relationship between signal activation and the number of motor vehicles that failed to stop in each lane of traffic during a given crossing episode. Negative binomial regression is used for the regression of outcomes that are count data, when the distribution is not well described by the Poisson distribution (we used it because the counts of motor vehicles that failed to stop was not Poisson distributed). This is now mentioned on page 6, lines 184-186.
5. What is the total number of intercept surveys?
122 intercept surveys were conducted. We have now mentioned this on page 8, line 224.
6. Please explain the difference between both descriptions of scoring of the motor vehicle behavior: lines 162-165 (correct stop=0 points) and lines 201-204 (correct stop =1 point).
Thank you for catching this. These statements were supposed to be describing the same scoring process, but the description in lines 162-165 was not updated to reflect changes made to the motor vehicle behavior score. 1 point was added to the score for each correct stop. The error in the methods has been corrected (page 5, lines 166-170).
Reviewer 3 Report
This is a very interesting paper, and it is well written. I have the following comments hoping to help the authors further improve the paper.
(1) I'd encourage the authors elaborate on the statistical analyses, especially the logistic and NB models' structure. For example, what are the equations linking response and independent variables in the two models?
In the NB model (Tab 5), shouldn't there be an intercept parameter? Please also report the dispersion parameter estimates of the NB models.
The parameter for pedestrian in all the three NB models are not statistically significant. Is there any possible reason? Please discuss.
(2) This paper focuses on the behavior of pedestrians, cyclists as well as drivers, which is good. From roadway agency (i.e., MPO, or DOT) perspective, we are more interested in the the actual crash reduction. Though it is beyond the scope of this paper, it might be good to mention that crash-based evaluation is needed in the future. Traffic safety analysts have proposed methods for conducting safety evaluation (usually known as Crash Modification Factor assessment). Just an example for your reference: https://doi.org/10.1177/0361198118776481
I enjoyed reading this paper.
Author Response
Reviewer 3:
General Comments:
This is a very interesting paper and it is well written. I have the following comments hoping to help the authors further improve the paper.
Thank you, we appreciate the thoughtful feedback.
Specific Comments:
1. I’d encourage the authors to elaborate on the statistical analyses, especially the logistic and NB models’ structure. For example, what are the equations linking response and independent variables in the two models?
We have provided additional information about the outcomes for logistic and negative binomial regression models, and brief justification of why we selected those types of regression models (pages 5-6, lines 181-186). We have not included the equations linking the response and independent variables for the models, because we feel that the equations will distract readers from the content of the manuscript. We have provided additional citations (page 6, lines 184, 186: Hosmer DW, Lemeshow S. Applied Logistic Regression, Third Edition. Hoboken, NJ: John Wiley & Sons; 2013. : Cameron AC, Trivedi PK. Regression Analysis of Count Data. New York: Cambridge Press; 1998 : Payne EH, Hardin JW, Egede LE, Ramakrishnan V, Selassie A, Gebregziabher M. Approaches for dealing with various sources of overdispersion in modeling count data: Scale adjustment versus modeling. Stat Methods Med Res. 2017 Aug;26(4):1802-23,doi.org/10.1177/0962280215588569 ) so readers who wish to learn more about the types of regression models may do so.
2. In the NB model (table 5), shouldn’t there be an intercept parameter? Please also report the dispersion parameter estimates of the NB models.
Yes, there is an intercept parameter in each negative binomial regression model, but as we were interested in reporting the association between signal use by cyclists/pedestrians and motor vehicle behavior, we have not reported these intercept estimates. Dispersion parameters were not included in the paper as they are not commonly reported (dispersion parameter estimates β(SE) 0.0000(0.0008), 0.0000(0.0000), and 0.0000(0.0007) for model 1, model 2, and model 3 respectively).
3. The parameter for pedestrian in all three NB models are not statistically significant. Is there any possible reason? Please discuss.
As we state on page 14, lines 325-334, we think there is no association between signal activation and motor vehicle yielding (RRFB activation has no impact on motor vehicle behavior). Since pedestrians are more likely to use the signal, the lack of an association between signal use and motor vehicle behavior is appropriately reflected for pedestrians. Cyclists, who may only use the signal when traffic is heavy, may have a biased estimate of the association (truly no association between signal activation and motor vehicle behavior, but appears to be less yielding with signal activation). The sample of crossing observations was predominantly cyclists, which is reflected in the smaller standard error estimates for negative binomial models for cyclist use of the signals compared to pedestrian use of the signals.
4. This paper focuses on the behavior of pedestrians, cyclists as well as drivers, which is good. From roadway agency (ie MPO or DOT) perspective, we are more interested in the actual crash reduction. Though it is beyond the scope of this paper, it might be good to mention that crash-based evaluation is needed in the future. Traffic safety analysts have proposed methods for conducting safety evaluation (usually known as Crash Modification Factor assessment).
Thank you, we agree that crash-based evaluation would be valuable and have added mention of this in the conclusions (Page 15, lines 368-369).
Reviewer 4 Report
GENERAL COMMENTS:
An interesting analysis of a greenway route, with some unexpected results. The resulting findings suggest to me that better street treatments would be warranted at these crossings, e.g. speed management, full signalized crossings.
- The abstract refers to "crossing signals", which in many jurisdictions means full 2-3 aspect displays ("traffic lights"). It is only in the body of the paper that we learn this study is looking at RRFBs, quite a different treatment and not really "signals". This should be made clear in the abstract, lest anyone gets a misleading impression from just reading the abstract.
- It is difficult to comment on the relative effectiveness of these RRFBs (in terms of activation rates and compliance), without comparison with other similar RRFB sites.
- There appears to be no specific analysis of whether activation or compliance or RRFBs is influenced by the presence or otherwise of traffic (esp. high vs low volume periods). I could understand why people wouldn't necessarily bother pushing the button if there is no traffic around, for example. Some comparison of the observed metrics vs (say) the hourly traffic volumes at the time would be useful to see.
- It would be helpful to see some photos of typical crossings studied (esp. from motorist point of view) to help unfamiliar readers understand the context better.
- The page numbering keeps resetting throughout the paper (usually after changes between portrait/landscape orientation); ensure that numbering is continuous throughout.
SPECIFIC COMMENTS:
- Line 57: Given the different travel speeds involved, comparing deaths per HOUR traveled is a far more useful metric for comparing different travel modes.
- Lines 108/112: It's not clear which of the two study dates came first - is there a 3 month or 9 month time difference? Either way, there is likely to be quite a difference in patterns between these two time periods, partly due to seasonal differences in populations/behaviors.
- Line 127-8: I'm not clear how a crossing is "unsafe" on the part of the ped'n/cyclist if it was the motorist who failed to stop as required?
- Line 197 (Table 1): I'm struggling to see the relevance of including some of the socio-economic characteristics of each neighborhood. Also, these factors don't necessarily correlate with the on-road/path behavior of those users using the arterial roads/paths have (likely) traveled from different areas.
Some of the traffic volumes stated seem very high for just two lanes of traffic (would be struggling to have spare capacity). Also surprising that the street with the lowest volume (by far) is the only one with 4 traffic lanes.
- Line 207 (Table 2): I'm not sure that I would consider a crossing maneuver as "errant" if no-one was around on the road at the time.
- Line 217: Why is "data not shown"?
- Line 234-5: I don't know whether there is any legal difference regarding the requirement to stop for a crossing pedestrian vs bicyclist, but is it possible that there is a different perception by many drivers of what the law requires for each?
- Line 238 (Table 3): Shouldn't split table across pages (esp. without repeating headings). What about other possible greenway trip purposes, e.g. utility trips to shops or visiting friends/family? Under "feel that drivers at crossings..." there must be some overlap in people responding with "drivers behave well" and "drivers speed up", which seems contradictory.
- Line 239: It's not obvious that this is a footnote to Table 3
- Line 249-250 (Table 4): I wonder whether some of this data is related to how quickly traffic can stop after the RRFB button is pressed? Path users may be crossing almost immediately after pressing, whereas motorists may need a few seconds to respond and to stop.
- Line 288-300: As you didn't undertake a true "before and after" study at your sites, it is difficult to know whether behavior has improved from previously.
- Line 321-322: My understanding is that you observed path and road user behavior at different times, thus making it more difficult to directly compare these.
Author Response
Reviewer 4:
General Comments:
An interesting analysis of a greenway route with some unexpected results. The resulting findings suggest to me that better street treatments would be warranted at these crossings, e.g. speed management, full signalized crossings.
Thank you. We appreciate your comments and agree that the crossings would benefit from modified treatments. We have mentioned some of the possibilities for better street treatments at the crossings in the conclusions (page 15, lines 371-374).
Specific Comments:
1. The abstract refers to “crossing signals” which in many jurisdictions means full 2-3 aspect displays (“traffic lights”). It is only in the body of the paper that we learn this study is looking at RRFBs, quite a different treatment and not really “signals”. This should be made clear in the abstract, lest anyone gets a misleading impression from just reading the abstract.
We have now clarified this in the abstract on page 1, lines 27 and 32.
2. It is difficult to comment on the relative effectiveness of these RRFBs (in terms of activation rates and compliance) without comparison with other similar RRFB sites.
We have added more information on activation rates and signal compliance with RRFBs in other cited studies in the Discussion at page 14, lines 308-317.
3. There appears to be no specific analysis of whether activation or compliance or RRFBs is influenced by the presence or otherwise of traffic (especially high vs low volume periods). I could understand why people wouldn’t necessarily bother pushing the button if there is no traffic around, for example. Some comparison of the observed metrics vs (say) the hourly traffic volumes at the times would be useful to see.
As we don’t have hourly traffic volume estimates for the streets in this study, we are unable to perform this analysis. Research assistants only recorded motor vehicle counts while the street crossings along the greenway were in use, and only motor vehicles that interacted with the crossing, and therefore the count of motor vehicles in a time period is a reflection of both the number of greenway users during that time period, the volume of motor vehicle traffic, and how many vehicles in each lane of traffic failed to stop before one stopped (vehicles in a lane behind the stopped vehicle were not counted).
4. It would be helpful to see some photos of typical crossings studies (especially from the motorist point of view) to help unfamiliar readers understand the context better.
We have added a Figure 2 (page 4), consisting of images showing one of the crossings in this study from the motorist perspective (Figure 2A) and greenway user (Figure 2B) perspectives.
5. The page numbering keeps resetting throughout the paper usually after changes between portrait/landscape orientation); ensure that numbering is continuous throughout.
Thank you, we have corrected this so that page numbering is continuous throughout the document.
6. Line 57: Given the different travel speeds involved, comparing deaths per HOUR traveled is a far more useful metric for comparing different travel modes.
We have added a statement that the risk of death measured per trip is higher for both cyclists and pedestrians compared to motor vehicle occupants on page 2 lines 57-58, however we feel that travel is done to move across distances. The value of a mile traveled may differ by mode of transportation, due to differences in speed of travel, and the distance of a trip may influence the selection of the mode of transportation, but travel is done to move across a physical, not temporal, distance.
7. Lines 108/112: It’s not clear which of the two study dates came first – is there a 3 month or 9 month time difference? Either way, there is likely to be quite a difference in patterns between these two time periods, partly due to seasonal differences in population/behaviors.
Both dates were in 2017. We have mentioned the year for each date of observations (page 4, lines 113 and 117). Both were Friday mornings, with no precipitation, when public schools were in session, without a local state or federal holidays, in a region with minimal seasonal temperature variation. We expect there to be very little difference in population or seasonality of behavior, and neither of these should alter the association between signal use, motor vehicle behavior and crossing safety. The data collected on the two days of direct observation were not combined for any part of the analysis, so the analysis is stratified by date of collection, which should control the influence of the time period from influencing the results.
8. Line 127-128: I’m not clear how a crossing is “unsafe” on the part of the pedestrian/cyclist if it was the motorist who failed to stop as required?
Unsafe is not intended as a criticism of the pedestrian or cyclist, but as a description of the crossing while they are in lanes of motor vehicle traffic. The crossings were unsafe in that they conferred a heightened risk to the pedestrian or cyclist, not necessarily because the cyclist or pedestrian did something that was reckless.
9. Line 197 (Table 1): I’m struggling to see the relevance of including some of the socio-economic characteristics of each neighborhood. Also, these factors don’t necessarily correlate with on-road/path behavior of those users using the arterial roads/paths have likely traveled from different areas.
This information was included because the introduction of the greenway was conceived as a way of improving these “deprived” neighborhoods, and this information provides context for readers. Additionally, this manuscript is being submitted for inclusion in a special issue of this journal on neighborhood influences on health.
10. Some of the traffic volumes stated (Table 1) seem very high for just two lanes of traffic (would be struggling to have spare capacity). Also surprising that the street with the lowest volume (by far) is the only one with 4 traffic lanes.
Some of these vehicle counts are from sections of the streets up to 0.5 miles away from the intersection with the greenway (we always selected the closest location with vehicle counts). At the locations where traffic volume was assessed, streets 1, 2, 5, 6, 8 and 9 all have 3 lanes of traffic.
11. Line 207 (Table 2): I’m not sure that I would consider a crossing maneuver as “errant” if no-one was around on the road at the time.
The assessment of behavior needs to be consistent across all observations, otherwise we might introduce an information bias to our study. Errant crossings may not be risky in the absence of motor vehicle traffic, but for the evaluation of the prevalence of errant behavior, it is necessary to consistently categorize behavior regardless of motor vehicle presence. From a legal perspective, jay walking and running red lights are illegal behavior regardless of whether any other people or vehicles are present, though they may be of no risk without the presence of other vehicles.
12. Line 217: Why is “data not shown”?
The data is not shown because small non-significant and unadjusted correlation coefficients were deemed less informative, as correlations may be difficult to interpret, than results of regression models that are more readily interpreted.
13. Line 234-235: I don’t know whether there is any legal difference regarding the requirement to stop for a crossing pedestrian vs bicyclist, but is it possible that there is a different perception by many drivers of what the law requires for each?
It is possible that drivers believe pedestrians are endowed with right of way and cyclists are not. As we were unable to conduct intercept interviews among drivers as a part of this study, we can’t make any assertion as to whether hypothesized driver perception underlies differences in signal response.
14. Line 238 (Table 3): Shouldn’t split the table across pages (especially without repeating headings). What about other possible greenway trip purposes (eg utility trips to shops or visiting friends/family)? Under “feel that drivers at crossings…” there must be some overlap in people responding with “drivers behave well” and “drivers speed up”, which seems contradictory.
Thank you, we have made sure that the table is contained in one page. While the responses to the survey may seem contradictory, we have reported the responses as given by the respondents. There is at least a subset of survey respondents who believe that drivers speed up in response to the signals and that drivers generally behave well at the crossings.
15. Line 239: It’s not obvious that this is a footnote to Table 3.
We have changed the formatting to make this more clear.
16. Line 249-250 (Table 4): I wonder whether some of this data is related to how quickly traffic can stop after the RRFB button is pressed? Path users may be crossing almost immediately after pressing, whereas motorists may need a few seconds to respond and to stop.
It is possible that some of the observed association is due to the inability of motor vehicle traffic to stop quickly enough, but this is still relevant to our evaluation of the effectiveness of the signals at making vehicles stop. If there is a disconnect between what the cyclists/pedestrians need and the ability of the drivers to conform to the needs of cyclists/pedestrians, then this disconnect needs to be considered during further intersection modifications.
17. Line 288-300: As you didn’t undertake a true “before and after” study at your sites, it is difficult to know whether behavior has improved from previously.
We have acknowledged this in the limitations of our study (page 15, lines 345-347), but would emphasize that if driver behavior doesn’t differ between when the signal is activated and is not activated, it is unlikely that much of the hypothetical improvement in driver behavior from before signals were installed would be attributable to the intended function of the signals.
18. Line 321-322: My understanding is that you observed path and road user behavior at different times, thus making it more difficult to directly compare these.
We observed both path and road user behavior on both days. On the second day of crossing observation, we did not record path user behavior beyond whether they used the signal to attempt a crossing. For the purpose of these statements in the discussion, we estimate the number of unsafe crossings in a day based on the daily average number of greenway users and the prevalence of unsafe crossings from the first day of direct observation.
Round 2
Reviewer 1 Report
Dear Authors,
Thank you for your work. The manuscript has been improved.